# Spironolactone as a Potential New Treatment to Prevent Arrhythmias in Arrhythmogenic Cardiomyopathy Cell Model

**DOI:** 10.3390/jpm13020335

**Published:** 2023-02-15

**Authors:** Jean-Baptiste Reisqs, Adrien Moreau, Yvonne Sleiman, Azzouz Charrabi, Antoine Delinière, Francis Bessière, Kevin Gardey, Sylvain Richard, Philippe Chevalier

**Affiliations:** 1Neuromyogene Institute, Claude Bernard University, Lyon 1, 69008 Villeurbanne, France; 2PhyMedExp, INSERM, University of Montpellier, CNRS, 34000 Montpellier, France; 3Service de Rythmologie, Hospices Civils de Lyon, 69500 Lyon, France

**Keywords:** arrhythmogenic cardiomyopathy, spironolactone, arrhythmias, hiPSC-CM

## Abstract

Arrhythmogenic cardiomyopathy (ACM) is a rare genetic disease associated with ventricular arrhythmias in patients. The occurrence of these arrhythmias is due to direct electrophysiological remodeling of the cardiomyocytes, namely a reduction in the action potential duration (APD) and a disturbance of Ca^2+^ homeostasis. Interestingly, spironolactone (SP), a mineralocorticoid receptor antagonist, is known to block K^+^ channels and may reduce arrhythmias. Here, we assess the direct effect of SP and its metabolite canrenoic acid (CA) in cardiomyocytes derived from human-induced pluripotent stem cells (hiPSC-CMs) of a patient bearing a missense mutation (c.394C>T) in the *DSC2* gene coding for desmocollin 2 and for the amino acid replacement of arginine by cysteine at position 132 (R132C). SP and CA corrected the APD in the muted cells (vs. the control) in linking to a normalization of the hERG and KCNQ1 K^+^ channel currents. In addition, SP and CA had a direct cellular effect on Ca^2+^ homeostasis. They reduced the amplitude and aberrant Ca^2+^ events. In conclusion, we show the direct beneficial effects of SP on the AP and Ca^2+^ homeostasis of *DSC2*-specific hiPSC-CMs. These results provide a rationale for a new therapeutical approach to tackle mechanical and electrical burdens in patients suffering from ACM.

## 1. Introduction

Arrhythmogenic cardiomyopathy (ACM) is a rare inherited heart condition, often linked to mutations in genes encoding desmosomal proteins [1,2] (OMIM 610476). ACM is characterized by progressive mechanical and electrical myocardial dysfunction [3,4]. In the concealed stage of this disease, there is no mechanical dysfunction but a risk of sudden cardiac death due cardiac arrhythmias, such as ventricular fibrillation [5,6]. How this disease predisposes patients to ventricular fibrillation is not fully understood [7,8]. In a transgenic mice model with a cardiac-specific overexpression of *DSC2*, severe myocardial necrosis, fibrosis, and inflammation mimicking ACM are found [9]. Recent studies on ACM patient-specific induced pluripotent stem cells (ACM-hiPSC-CMs) have shown that profound electrophysiological remodeling of cardiomyocytes contributes to arrhythmias [4]. Recently, a thorough analysis of hiPSC-CMs derived from a patient with a heterozygous missense mutation (c.394C>T) in exon 3 of the *DSC2* gene (NM_024422.6:c.394C>T[p.Arg132Cys], which encodes desmocollin 2, revealed abnormal cellular early repolarization dynamics in line with a short QT interval in some ACM patients [10]. This observation is one of the major causes of triggering and maintaining ventricular arrhythmia in ACM patients and increasing dispersion repolarization, which is a pro-arrhythmogenic mechanism [11]. Normalizing the QT interval can be an interesting target to prevent arrhythmia in ACM patients.

For a long time, many studies have shown that angiotensin-converting enzyme inhibitors are associated with a lower incidence of ventricular arrhythmias in patients with heart failure or LV dysfunction [12,13]. In the Heart Failure Efficacy and Survival Study (EPHESUS) and Randomized Aldactone Evaluation Study (RALES), mineralocorticoid receptor antagonists (MRAs) reduced the risk of ventricular arrhythmias [14]. A direct anti-arrhythmic effect of MRAs has been suggested by several studies [15,16,17]. A phase II clinical study has been initiated in patients with ACM to investigate the impact of spironolactone (SP) on arrhythmias [18]. Interestingly, SP is known to block K^+^ channels [19]. Our work aims to assess the direct effects of SP and its metabolite, canrenoic acid (CA), on patient-specific human *DSC2* mutated hiPSC-CMs (ACM-DSC2-hiPSC-CMs) to assess the potential benefits to prevent or reduce ACM-induced electrical instability.

## 2. Methods

### 2.1. Clinical Patient Case

The patient, in the clinical case, harbored two mutations of the *DSP* gene. One was on exon 23, inducing a heterozygous *DSP* mutation (c.4565C>T) and coding for an amino acid replacement of threonine by methionine at position 1522 (T1522M). The second was a heterozygous deletion of exon 24. The deletion of this exon is a new variation whose implication cannot be affirmed with certainty but is highly probable. The search for the deletion was carried out by the MLPA method (kit P168, MRC Holland).

### 2.2. Cell Culture

The methods are described elsewhere [10]. Briefly, we used two control hiPSC lines, one from the platform IPS from INMG (hiPSC AG08C5; Neuromyogen institute, Lyon, France) and the other from Dr. Xavier Nissan (hiPSC FSC03; I-Stem institute, Evry, France). We cultured the hiPSC with StemFlex^TM^ in a DMEM/F12 medium for 5 days (ThermoFisher Scientific, Waltham, MA, USA) and differentiated them using the sandwich monolayer technique. We induced cardiac differentiation by adding 6 µM of CHIR99021 (LC Laboratories, MA, USA) and then 2 µM of Wnt-C59 (LC Laboratories) for each molecule for 2 days. Previously, we showed that the DSC2 phenotype is best expressed at 60 days (D60) [10]. Finally, for “single-cell” experiments, we dissociated the cardiomyocytes at D53 using TrypLE^TM^ Express (ThermoFisher Scientific) and replated them on Matrigel-coated 35 mm Petri dishes for patch-clamp experiments. We incubated cells with SP at 20 μM or canrenoic acid at 10 nM for one hour before performing the experiments.

### 2.3. Whole-Cell Current-Clamp Recordings

For the current-clamp studies, we recorded the action potential (AP) and global K^+^ currents in single cells at room temperature (22 ± 2 °C) using an Axopatch 200B amplifier (Axon Instruments, Foster City, CA, USA). The resistance of patch pipettes ranged between 1 and 2 mΩ. We generated voltage-clamp command pulses using pCLAMP v10 (Axon Instruments). We filtered currents at 5 kHz, digitized them at 10 kHz, and stored them on a microcomputer equipped with an AD converter (Digidata 1440A, Axon Instruments). The intracellular solution in the patch pipette for AP and K^+^ currents recordings contained (in mmol/L) the following: 10 NaCl; 122 KCl; 1 MgCl_2_; 1 EGTA; and 10 HEPES. Furthermore, the pH was adjusted at 7.3 with KOH. The bath solution contained (in mmol/L) the following: 154 NaCl; 5.6 KCl; 2 CaCl_2_; 1 MgCl_2_; 8 Glucose; and 10 HEPES. The pH was adjusted at 7.4 with 1N NaOH.

### 2.4. Calcium Handling in hiPSC-CMs

To measure spontaneous calcium (Ca^2+^) transients, we incubated hiPSC-CMs with 3 μM of Fluo-4 AM (Molecular Probes, Em/Ex: 494/506 nm) in a classic Tyrode solution (154 mM NaCl; 5.6 mM KCl; 2 mM CaCl2; 1 mM MgCl2; 8 mM Glucose; 10 mM HEPES; pH 7.4). We recorded fluorescence images using confocal microscopy (Zeiss LSM 810) in line-scan mode (i.e., x–t mode, 1.24 ms per line; 512 pixels × 5000 lines). To enable comparisons between cells, changes in the Fluo-4 fluorescence signal (F) were normalized by basal fluorescence (F0). We extracted all data using AIM 4.2 (Zeiss, Jena, Germany). We extracted maximal amplitudes and event frequencies and analyzed them from raw data through an in-house-developed algorithm implemented in Python.

### 2.5. Quantitative Reverse Transcription PCR (RT-qPCR)

Total RNA was isolated from hiPSC-CMs at 60 days using Direct-Zol RNA Miniprep, including DNAse treatment following the protocol (Zymo Research, Irvine, USA, ref: R2051). We performed reverse transcription of 500 ng total RNA into cDNA using the Transcriptor Universal cDNA Master kit (Roche, Basel, Switzerland, ref: 05893151001), following the manufacturer’s protocols. We carried out qPCR using LightCycler 480 SYBR Green I Master (Roche, ref: 04887392001), executed in quintuplicate. We used Ribosomal Protein Lateral Stalk Subunit P0 (RPLPO) as the internal control, and we normalized the mRNA levels for each sample. The primers for the interest genes are listed in Table 1.

### 2.6. Contraction of hiPSC-CMs’

We investigated the contraction of hiPSC-CMs in monolayer six-well dishes placed on the stage of an inverted microscope with a 10× objective and an incubation chamber (37 °C, 5% CO_2_). We acquired videos at 30 frames per second and performed analyses using custom-made Matlab scripts. We considered several contractile parameters as follows: the beat rate; the contraction and relaxation time; the amplitude of the contraction and area under the curve (AUC); and the summation of the contraction and the relaxation time. We also studied the asynchronous time, which is the time-fraction of each video area spent in asynchrony, and described the aberrant contractile events.

### 2.7. Data Analysis and Statistics

We analyzed the electrophysiological results using Clampfit (pCLAMP v10.0; Molecular Devices). We tested normality (Agostino test) to determine whether data followed a normal distribution. We expressed data with the median and a 95% confidence interval. We used the Kruskal–Wallis test to evaluate statistical differences and the stars * correspond to the difference from the control. We considered the differences significant at *p* < 0.05 (*), *p* < 0.01 (**), *p* < 0.001 (***), or *p* < 0.0001 (****).

## 3. Results

### 3.1. Aldactone Decreases Arrhythmic Events in an ACM Patient

We prescribed SP (50 mg/day) to an ACM patient harboring a desmoplakin (DSP) variant. Figure 1 shows ECGs and the number of premature ventricular beats (PVBs) before and after this prescription. We measured the QTc durations initially at 406 milliseconds (ms), and the patient experienced a significant number of PVBs (1.7%; Figure 1, panel A). After seven days of SP therapy, the QTc interval increased to 436 ms and the number of PVBs decreased to 0.8% (Figure 1, panel B). Based on the available literature, this finding stimulated our investigation into SP impact at the cardiomyocyte level. To test the hypothesis of a direct effect of aldosterone antagonists in ACM patients experiencing arrhythmias, we used hiPSC-CMs obtained from an ACM patient affected by a missense variant (c.394C>T) in the *DSC2* gene encoding desmocollin 2 (DSC2-hiPSC-CMs) [10,20].

### 3.2. SP and CA Normalize the AP and Reduce Electrical Instability in ACM-DSC2-hiPSC-CMs

We examined the cellular electrical activity using the patch-clamp technique. Spontaneous APs in the control hiPSC-CMs, ACM-DSC2-hiPSC-CMs, and after one-hour incubation of ACM-DSC2-hiPSC-CMs in the presence of SP (20 μM) and CA (10 nM) were recorded. We observed a difference in the firing of spontaneous APs with an increase in the frequency of APs per time unit between the control hiPSC-CMs (0.33 ± 0.06 Hz) and the ACM-DSC2-hiPSC-CMs (0.51 ± 0.07 Hz) (Figure 2A,B). AP duration (APD) at 90% of repolarization (APD_90_) was shorter in the ACM-DSC2-hiPSC-CMs (224 ± 32 ms) than in the control hiPSC-CMs (432 ± 43 ms) (Figure 2C) in line with previous results [10,20]. These alterations were similarly normalized following the acute application of SP or CA. Strikingly, we observed no difference in spontaneous AP frequency between the control hiPSC-CMs (0.33 ± 0.06 Hz) and the ACM-DSC2-hiPSC-CMs in the presence of SP (0.30 ± 0.03 Hz) or CA (0.40 ± 0.05 Hz) (Figure 2A,B). The APD normalized after the incubation of SP and CA in ACM-DSC2-hiPSC-CMs (471 ± 31 ms and 360 ± 29 ms for SP and CA, respectively, vs. 432 ± 43 ms for the control cells). We also measured the APD_90_ after pacing at 1 Hz and observed similar results (Appendix A).

The percentage of cells with aberrant electrical events, similar to early or delayed-after depolarization (EAD, DAD), was higher in the ACM-DSC2-hiPSC-CMs than in the control hiPSC-CMs (47% vs. 23%, respectively) (Figure 2D). Again, SP and CA attenuated the electrical instability of the ACM-DSC2-hiPSC-CMs. The number of ACM-DSC2-hiPSC-CMs presenting ectopic electrical events decreased to 30% and 27% with exposure to, respectively, SP and CA (Figure 2D).

We assessed the direct effect of CA and SP on the global outward K^+^ current using the patch-clamp technique. ACM-DSC2-hiPSC-CMs exhibited a larger K^+^ current than the control hiPSC-CMs (Figure 3). Acute exposure of ACM-DSC2-hiPSC-CMs to SP or CA decreased the K^+^ currents. The K^+^ currents were normalized for voltages ranging between −40 mV and 0 mV on one hand and further reduced between 0 and +40 mV with an amplitude lower than that of the control hiPSC-CMs (Figure 3A–E). For example, at +40 mV, the K^+^ currents were lower in ACM-DSC2-hiPSC-CMs in the presence of SP and CA (0.9 ± 0.1 pA/pF and 1.2 ± 0.1 pA/pF, respectively) than in the control hiPSC-CMs (2.3 ± 0.3 pA/pF) (Figure 3E). RT-qPCR analysis of the K^+^ channels potentially involved suggested the presence of hERG and KCNQ1 proteins in hiPSC-CMs and unraveled an increase in their expression in ACM-DSC2-hiPSC-CMs compared to the control hiPSC-CMs (Figure 3F).

### 3.3. SP, CA, and Eplerenone Normalize Ca^2+^ Cycling in ACM-DSC2-hiPSC-CMs

We investigated Ca^2+^ cycling with the fluorescent Ca^2+^ dye fluo-4 AM. We observed distinct Ca^2+^ transient waveforms in the control hiPSC-CMs and the ACM-DSC2-hiPSC-CMs. The amplitude was reduced in the ACM-DSC2-hiPSC-CMs compared to the control hiPSC-CMs (1.7 ± 0.1 F/Fo vs. 2.8 ± 0.1 F/Fo, respectively). The area under the curve (AUC) followed a similar trend (2.9 ± 0.2 × 10^6^ A.U. vs. 4.0 ± 0.6 × 10^6^A.U.) (Figure 4A–F). Differences between the ACM-DSC2-hiPSC-CMs and the control hiPSC-CMs were abolished following acute exposure of ACM-DSC2-hiPSC-CMs to SP (amplitude: 2.7 ± 0.1 F/Fo; AUC: 3.5 ± 0.2 × 10^6^ A.U.) and CA (amplitude: 2.5 ± 0.1 F/Fo; AUC 3.3 ± 0.3 × 10^6^ A.U.) (Figure 4E,F), which both normalized these parameters. The number of cells presenting aberrant Ca^2+^ events was also higher in the ACM-DSC2-hiPSC-CMs than in the control hiPSC-CMs (41% vs. 19%, respectively) (Figure 4G). Again, SP and CA normalized the occurrence of these Ca^2+^ events (22% and 20%, respectively) (Figure 4G). Interestingly, these two compounds prevent microscopic aberrant calcium events, i.e., Ca^2+^ sparks (Figure 5). Of note, we tested eplerenone (500 nM), another aldosterone inhibitor specific to mineralocorticoid receptors, and we found similar results (Appendix A).

### 3.4. Effect of CA on Contractile Properties in ACM-DSC2-hiPSC-CMs

We assessed the contractile function of hiPSC-CMs from the cell monolayer after 60 days of differentiation using video-edge capture [10]. The contractile activity was disturbed with an increase in the asynchrony rate of ACM-DSC2-hiPSC-CMs compared to the control hiPSC-CMs (Figure 6). After one hour of incubation with CA, the beating rates and the amplitude of contraction were normalized compared to the control hiPSC-CMs (Figure 6A,B). Various parameters, such as contraction, relaxation duration, and the AUC (representing all contractile parameters), were not significantly affected by CA, despite a trend in normalization, for example, for relaxation duration and AUC (Figure 6C–E). CA did not change the asynchronous rate (Figure 6F). These results show that CA was less successful in rescuing contractile defects than the proarrhythmogenic electrical and Ca^2+^ handling abnormalities in ACM-DSC2-hiPSC-CMs.

## 4. Discussion

This study documents, for the first time, the acute electrophysiological effects of the prototypic MR antagonist SP and its metabolite CA on cardiomyocytes derived from an ACM patient. Firstly, we produced a model of ACM using human-induced pluripotent stem cells, which describe the hallmark of this pathology. In fact, previous studies have suggested that the origin of arrhythmia in ACM is not exclusively due to morphological remodeling [4,21]. The myogenic origin of electrical instability was recently suggested in ACM-DSC2-derived hiPSC-CMs with alterations of both Na^+^ and K^+^ currents, as well as Ca^2+^ dynamics [10,20]. In this model, profound electrophysiological remodeling with a critical APD shortening reflects a large increase in repolarizing K^+^ currents. This APD shortening strengthens the concept that ACM involves a disorder of repolarization and fits with our previous observations of short QT intervals in ACM patients [22]. hiPSCs allow us to discover novel potential treatments to prevent arrhythmias in cardiac pathology.

The RALES (Randomized Aldactone Evaluation Study) showed that SP added to ACE inhibitor therapy reduces the risk of sudden cardiac death (SCD) in patients with severe heart failure and a reduced left-ventricular ejection fraction [15,23]. The effect of SP in the RALES is based on cortisol-occupied (and activated) MR in stressed cardiomyocytes [15]. SP antagonizes the effect of aldosterone and can thereby reduce fibrosis and improve left-ventricular (LV) function. To our knowledge, SP has never been tested in ACM patients. ACM is highly arrhythmogenic and a significant cause of SCD in the young, particularly in athletes [24]. We have now established the acute favorable direct effects of SP, CA, and eplerenone on the cellular electrical activity and Ca^2+^ cycling in ACM-DSC2-hiPSC-CMs. SP and CA normalized AP duration and suppressed abnormal and irregular Ca^2+^ events, which are expected to decrease pro-arrhythmogenic risks of myogenic origin in ACM disease, independently of potentially favorable long-term effects on cardiac structural remodeling during disease progression.

A key result of our present study is that SP and CA acutely inhibited the global outward K^+^ current which, as an immediate consequence, normalized both the short APD and the firing of spontaneous APs in ACM-DSC2-derived hiPSC-CMs. In the human myocardium, the APD is controlled by several outward K^+^ channels, notably I_Kr_ (Kv11.1 protein encoded by the *KCNH2* gene) and I_Ks_ (Kv7.1 protein encoded by the *KCNQ1* gene), which we found to be overexpressed, in line with AP shortening, in our ACM-DSC2-derived hiPSC-CMs. These channels are important determinants of AP repolarization in the human ventricle and pace-making activity [25,26]. Of major importance is that the effects reported here are not attributable to aldosterone receptor antagonism as suggested before for HERG currents [19,27]. Indeed, SP and its metabolite CA can directly block the HERG channels expressed in stably transfected Chinese hamster ovary cells, i.e., in the absence of aldosterone, in addition to suppressing the native I_Kr_ in single guinea-pig ventricular myocytes [19]. Of note, SP and CA can target these channels in a voltage- and frequency-independent manner, preferentially in the closed and open states, according to the modulated receptor hypothesis [19,25,28]. This aligns with our observed effects of K^+^ current normalization between −40 mV and 0 mV, coinciding with channel activation, and further inhibition above 0 mV (i.e., when channels are fully activated) in the ACM-DSC2-derived hiPSC-CMs. We previously characterized the presence of the I_Kr_ current in our cellular model of ACM by a specific blocker of this current E4031 [10,29]. We found prolonging effects on the APD [10] very similar to the prolonging effect of SP shown here. We did not investigate the effect of SP on the I_to_ current because hiPSC-CMs do not express the protein responsible for this current [30]. However, a blocking effect has previously been reported in mice [27], which could contribute to the effect of SP in patients These direct effects, modulated by voltage and pacing rate, may beneficially modify vulnerable parameters, such as the normalization of AP duration, to stop or prevent arrhythmias [31].

Abnormalities of Ca^2+^ handling and enhanced pro-arrhythmogenic microscopic and macroscopic irregular or ectopic Ca^2+^ events are known to increase cellular excitability in a variety of cardiac diseases [32]. These trigger DADs and spontaneous APs and increase the occurrence of EADs as well, increasing cellular excitability [33,34,35]. Another important finding of our results is that SP, CA, and eplerenone normalized Ca^2+^ handling and limited the occurrence of EADs and DADs in ACM-DSC2-hiPSC-CMs. This result may be in line with a study showing that the main metabolite of SP, CA, decreased the beat rate by blocking the L-type calcium current in rabbit cardiomyocytes [36,37]. Our finding of an acute, direct effect of SP on Ca^2+^ events may also be reminiscent of a recent report showing the enhancing effect of short-term treatment with aldosterone on Ca^2+^ spark frequency in vascular myocytes from mesenteric arteries [38]. Because SP had effects on targets other than aldosterone receptors, proteins involved in Ca^2+^ homeostasis may be implicated. The ryanodine receptor (RYR) is a likely candidate because Ca^2+^ sparks reveal the activity of this receptor, and this hypothesis warrants further investigation. Interestingly, aldosterone has multiple cardiotoxic effects, including oxidation of CaMKII [39], which is expected to promote leakage through RyR2 with an increased occurrence of Ca^2+^ sparks and proarrhythmogenic Ca^2+^ waves [40,41].

Overactivation of the RAA system promotes cardiac structural remodeling (dilatation, fibrosis, and hypertrophy), which impairs impulse propagation and promotes reentrant arrhythmias [15,25,42]. The use of an aldosterone antagonist, such as SP, combined with an ACE inhibitor thus allows a dual blockade of the RAA [43]. SP slows the development of ventricular electrophysiological remodeling due to myocardial interstitial fibrosis and inflammation. It lowers arrhythmia inducibility in systolic heart failure. It is unclear whether SP or CA can potentially delay cardiac remodeling and slow the progression of the disease in ACM but their direct effects on K^+^ currents and Ca^2+^ cycling have excellent potential to prevent arrhythmias in ACM patients. Consistently, an ACM patient given SP for 2 weeks experienced fewer VPBs. This patient harbored a mutation of DSP, which may limit the conclusions of our study. Furthermore, this patient does not exhibit a short QT interval, as previously described by Chevalier et al. in 2021, as this is only visible at low heart rates [22]. Unfortunately, we did not obtain an ECG recording from this patient at a low heart rate. However, in a group of ACM patients with different desmosomal mutations, we showed a similar ECG phenotype with a QTc shortening [10]. Other studies in ACM-hiPSC-CMs with different desmosomal mutations also described similar cellular electrophysiological remodeling [3,4,44]. Altogether, these results strengthen the use of SP as an antiarrhythmic in ACM patients, irrespective of the underlying mutation, which is shown in Figure 7. A new randomized clinical study comparing SP vs. placebo in ACM patients is ongoing (BRAVE) [18]. SP could provide a new treatment to reduce electrical complications in patients suffering from ACM, a devastating disease for which there is no specific therapy. The results of the randomized BRAVE study are eagerly awaited.

Of note, in our investigations of contractile activity, CA failed to reduce the rate of asynchrony, which is the time-fraction of each video area spent in asynchrony. There is a gap between the beneficial effect observed in patients and at the cellular level and in the monolayer of several cells, despite a tendency to normalize certain parameters of the contraction. A possible explanation is that the immaturity of hiPSC-CMs would mask the beneficial effect of the drugs at the monolayer level. Moreover, we derived ACM-hiPSC-CMs from one patient with a *DSC2* mutation. It would have been interesting to extend our research with other hiPSCs carrying another mutation.

## 5. Conclusions

In summary, our cells models, based on hiPSC-CM, suggest that spironolactone prevents electrophysiological and calcium-handling disturbances in ACM. These effects are mediated by the normalization of action potential duration by inhibiting the K^+^ currents and contributing to the normalization of calcium handling. In this study, we show the link between a clinical case and the use of human-induced pluripotent stem cells-derived cardiomyocytes. This present study describes a good example of using hiPSCs to modelize a disease and to find a new treatment in a clinical context.

## Figures and Tables

**Figure 1 jpm-13-00335-f001:**
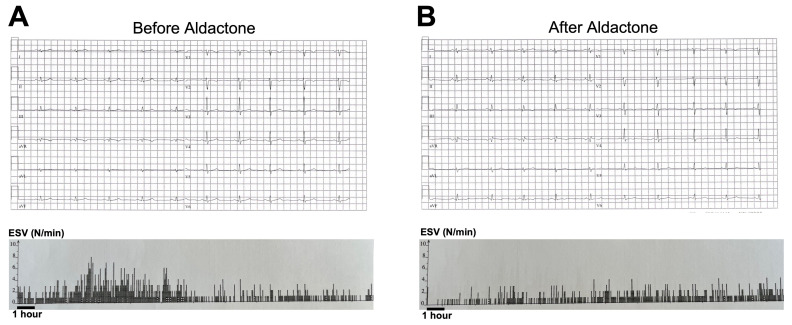
ECG after prescription of Aldactone in one ACM patient. (**A**) Example of ECG and PVB recording before Aldactone and (**B**) after Aldactone in a mutated desmoplakin (DSP) patient.

**Figure 2 jpm-13-00335-f002:**
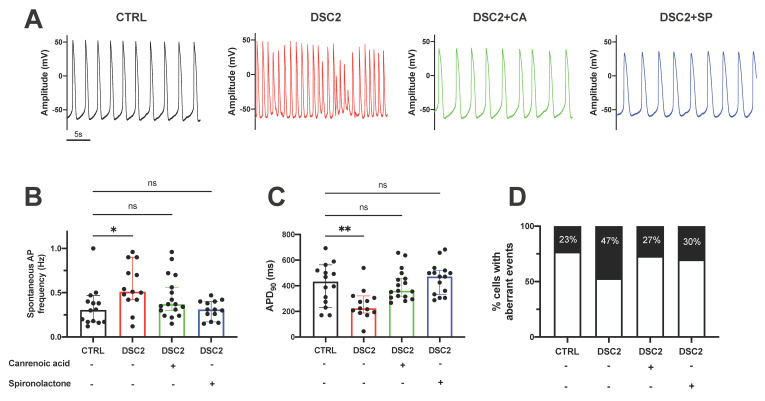
Effect of Canrenoic Acid and Spironolactone on the Action Potential parameters of DSC2 hiPSC−CM. (**A**) Raw traces of spontaneous AP in the control condition (black) (*n* = 14), mutated DSC2 (red) (*n* = 13), after incubation of 10 nM CA (green) (*n* = 18), and after incubation of 20 μM SP (blue) (*n* = 15). (**B**) Values of spontaneous frequency and (**C**) values of action potential duration at 90% repolarization. (**D**) Percentage of the number of cells presenting aberrant electrical events. Results are shown with the median and 95% confidence interval. The stars * correspond to the difference from the control. ns not statically significant, * *p* < 0.05, ** *p* < 0.01 (Kruskal–Wallis test).

**Figure 3 jpm-13-00335-f003:**
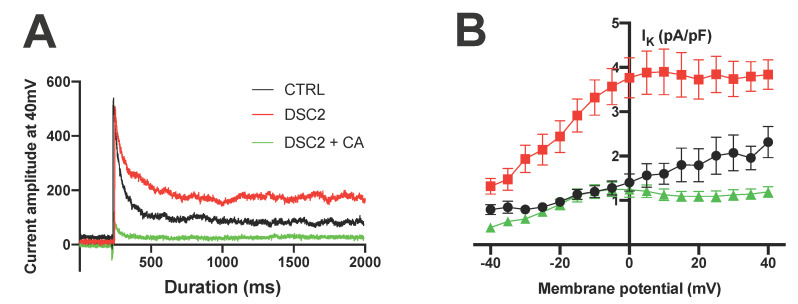
Effect of Canrenoic Acid on Potassium Channels. (**A**) Raw traces of K^+^ currents at +40 mV in control (black), DSC2 (red), and DSC2 + CA (green) condition and (**B**) potential current curves corresponding (CTRL *n* = 12; DSC2 *n* = 15; DSC2 + CA *n* = 10). (**C**) Raw traces of K^+^ currents at +40 mV in control (black), DSC2 (red), and DSC2 + SP (blue) condition and (**D**) potential current curves corresponding (CTRL *n* = 12; DSC2 *n* = 15; DSC2 + SP *n* = 11). (**E**) Values of Current amplitude at 40 mV for the different groups. (**F**) RT-qPCR results of fold change expression for *hERG* and *KCNQ1* gene in CTRL and DSC2 hiPSC−CM. The results are represented with the median and 95% confidence interval. The stars * correspond to the difference from the control. ns not statically significant, * *p* < 0.05, ** *p* < 0.01 (Kruskal–Wallis test).

**Figure 4 jpm-13-00335-f004:**
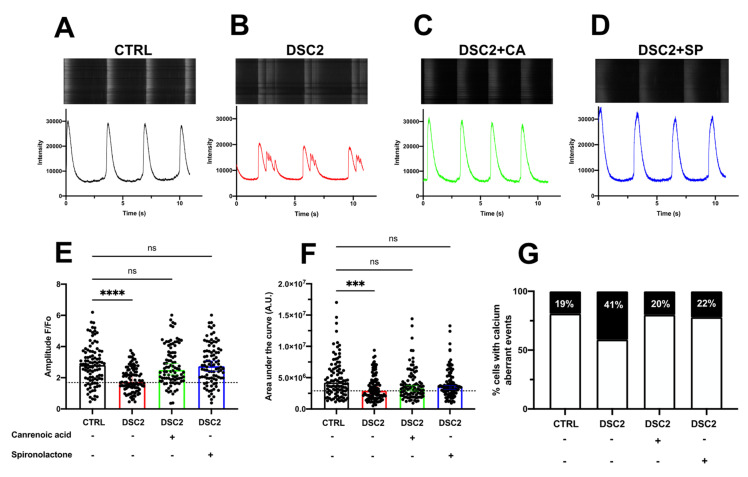
Effect of Canrenoic Acid on Calcium handling. Examples of spontaneous Ca^2+^ transient recorded in control (**A**) (black; *n* = 100), DSC2 (**B**) (red; *n* = 101), DSC2 + CA (**C**) (green; *n* = 92), and DSC2 + SP (**D**) (blue; *n* = 94). Values of amplitude (**E**,**F**), the area under the curve (AUC) of Ca^2+^ transients, and (**G**) percentage of cells presenting Ca^2+^ aberrant events for the different conditions. Results are represented with the median and 95% confidence interval. The stars * correspond to the difference from the control. ns not statically significant, *** *p* < 0.001 **** *p* < 0.0001 (Kruskal–Wallis test).

**Figure 5 jpm-13-00335-f005:**
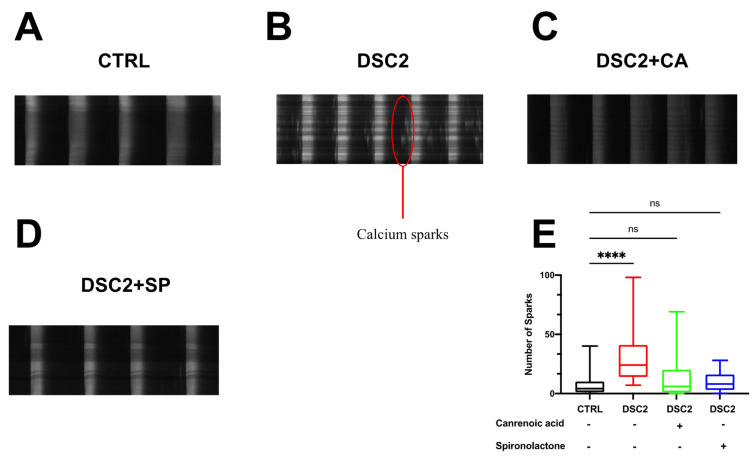
Effect of Canrenoic acid and Spironolactone on calcium sparks. Examples of spontaneous Ca^2+^ transient recorded showing the presence of calcium sparks in control (**A**), DSC2 (**B**), DSC2 + CA (**C**), and DSC2 + SP (**D**). (**E**) Measure of number of sparks in each condition. Results are shown by boxplot with the median. The stars * correspond to the difference from the control. ns not statically significant, **** *p* < 0.0001 (Kruskal–Wallis test).

**Figure 6 jpm-13-00335-f006:**
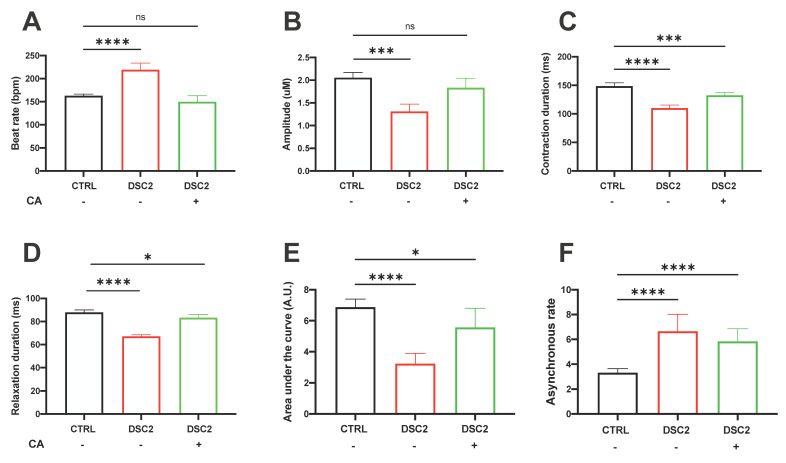
Effect of Canrenoic Acid on the Contractile Properties of hiPSC−CM. The contractile activity was studied by video analyses from a cell monolayer in control (black) (*n* = 386), patient (red) (*n* = 275) and incubated with AC (green) (*n* = 166): beat rate (**A**), amplitude (**B**), contraction duration (**C**), relaxation duration (**D**), area under the curve (**E**) and asynchronous rate (**F**). Results are shown with the median and 95% confidence interval. The stars * correspond to the difference from the control. ns not statically significant, * *p* < 0.05, *** *p* < 0.001 **** *p* < 0.0001 (Kruskal–Wallis test).

**Figure 7 jpm-13-00335-f007:**
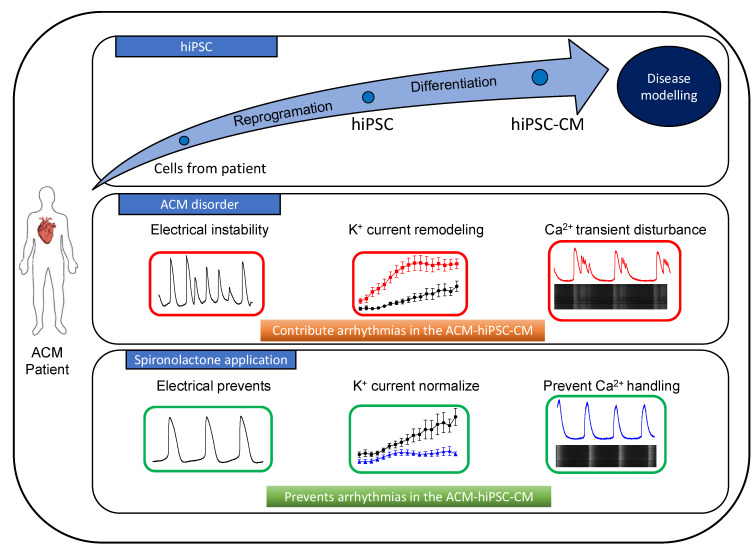
Schematic diagram of a benefits effects of Spironolactone on ACM-hiPSC-CM. HiPSC-CMs harboring a desmosomal mutation allow one to decipher the ACM disease. The electrophysiological and calcium-handling properties disturbed in this pathology. Application of Spironolactone prevents these effects, by a normalization of Action potential duration, electrical stability and a calcium handling restored. Spironolactone can reduce the risk of ventricular arrhythmias in ACM patients.

**Table 1 jpm-13-00335-t001:** Summary of genes evaluated in this study. Target gene, accession numbers, gene description, primer, and product sizes.

Target Gene	Accession Numbers	Gene Description	Primer Sequence (5’-3’)	Size (pb)
RPLP0	NM_001002.4	Ribosomal Protein Lateral Stalk Subunit P0	Forward: AGCAAGTGGGAAGGT Revers: TCATCCAGCAGGTGT	114
KCNQ1	NM_000218.3	K_v_7.1 (I_KS_)	Forward: CTCACTCATTCAGACCGCA Reverse: TCTTTACCACAGACTTCTTG	143
KCNH2	NM_000238.4	K_v_11.1 (I_Kr_)	Forward: GGAGCCTCTGAACCTGTATG Reverse: GACTGAAGCCACCCTCTAAC	230

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
