# Peer review of "Spironolactone as a Potential New Treatment to Prevent Arrhythmias in Arrhythmogenic Cardiomyopathy Cell Model"

_jpm, 2023, doi:10.3390/jpm13020335_

Round 1
Reviewer 1 Report
In the manuscript 'Spironolactone as a potential new treatment to prevent Arrhythmias in Arrhythmogenic Cardiomyopathy cell model', submitted by Reisqs JB to The Journal of Personalized Medicine (JPM), the authors investigate the molecular affects of Spironolactone on the contraction / electrophysiology of iPSC-derived cardiomyocytes carrying a DSC2 missense mutation. The manuscript is of general interests but needs several changes, before it should be accepted for publication in JPM. In the following these points are discussed in detail:
1.) All gene names should be written in Italics. Please add a OMIM number for ARCV/ACM within the introduction.
2.) Line 25: You should use the termin 'ventricular arrhythmias'
3.) Line 32: Please add the amino acid change nomenclature for this mutation.
4.) Line 52: Please list in addition primary literature for example Brodehl et al. 2020, JMCC, PMID 32201174).
5.) Line 59: Is the mutation homozygous or heterozygous?
6.) You should incorporate the information about DSC2 mouse models in your introduction including relevant citations.
7.) Please specifiy the timepoints (Line 85/Line 86)
8.) Please explain why room temperature was used and not 37°C (Line 94)?
9.) Please use capitals for HEPES (Line 102).
10.) Please specify the DSP mutation of the patient? How was the genetic analysis done? Please explain in the material method section? Is a pedigree available?
11.) Please correct all figures. Sometimes the stars indicating the p-values are missing (e.g. Figure 2).
However, I am optimistic that my criticized points can be changed in a minor revision. Good luck!
Author Response
We thank you for your time and correction of our work, and your constructive comments. We have taken your concerns and questions into consideration, and modified the text accordingly as detailed below.
Firstly, we added the supply information that you requested. In more detail, we have corrected the term ventricular arrhythmias (line 25) and added the amino acid change nomenclature R132C in the abstract. In the introduction part, we added and provided other information according to your comments. We added the OMIM number for ACM (OMIM number 610476) (line 53). The DSC2 gene mutation is heterozygous. We added this information (line 62). Finally, we listed additional literature and information about DSC2 mouse models (lines 53 and 57-59 respectively). All gene names have been written in italics throughout the manuscript. We have added the time points for the differentiation protocol. We have also written the word HEPES in capital letters in the material part.
Another issue concerned the use of room temperature (and not 37°C) for electrophysiological studies (line 94). It is true that physiological temperature is desirable as much as possible. However, room temperature has been currently used for electrophysiological investigations in numerous studies, mostly for technical reasons. For example, raising the temperature from ~20-22°C to 32-37°C has a major impact on several parameters like cell excitability, channel kinetics, and other biophysical indicators. High temperatures also increase the amplitude of ionic currents sensitive to metabolism, which hampers dramatically the technical quality of the recordings (significant measurement errors of the properties such as shifts in the voltage dependence, increased amplitudes, and faster kinetics), thereby introducing major errors in interpreting the experiments. Moreover, lower temperatures prevent excessive cell embrittlement and death during the course of experiments. In summary, the importance of temperature is a relative term and pertains to what you trying to measure and compare to answer your question. Many laboratories using this technique at room temperature have very strong results and the reproducibility of those results from one laboratory to another is very similar.
Please specify the DSP mutation of the patient? How was the genetic analysis done? Please explain in the material method section? Is a pedigree available?
The patient, in the clinical case, harbors two mutations of the DSP gene. One on exon 23 inducing a heterozygous DSPmutation (c.4565C>T), coding for an amino acid replacement of Threonine by Methionine at position 1522 (T1522M). The second is a heterozygous deletion of exon 24. The deletion of this exon is a new variation whose implication cannot be affirmed with certainty but is very probable. The search for the deletion was carried out by the MLPA method (kit P168, MRC Holland).
About your last comment, maybe there was a computer or compatibility problem because all the stars indicating significance are present. For example, in figure 2B, we compare the values with the control, and I see 1 star between ctrl vs. DSC2; ns between control vs. CA, and between control vs. SP. Same for figure 2C, with 2 stars between control vs. DSC2. Maybe I misunderstood your comment and if so, can you explain it to me in more detail?
We hope to have met your expectations and we remain available for another revision if necessary.
Reviewer 2 Report
The study examine the effects of spironolactone (SP), a mineralocorticoid receptor blocker, and its metabolite CA, on the cellular electrical properties and mechanical properties of hiPSC-CMs from a ACM patient with a mutation of the DSC2 gene, and showed that both compounds lengthened APD, likely through blockade of the K current, suppressed triggered activity-like aberrant firing, restored Ca transients to normal, suppressed abnormal Ca events, and finally, restored myocyte contractility. While interesting the study felt superficial and incomplete. I have the following concerns and questions.
The authors stated that the electrical abnormality of ACM patients involves short QT, and indeed, the DSC2 cells that the authors examined indeed showed shortened APD. However, the patient that was examined in the study had a QTc of about 400 ms, which was well within the the normal QTc range. Please comment on this. Also, the authors stated that the arrhythmias in ACM patients are at least in part due to electrical remodelings of the heart, which indicates that the electrical abnormalities of ACM myocytes are not entirely genetic based. This raises the question of whether hiPSC-CMs, which carry the genetic mutations but lack the remodelings, are a suitable model of ACM.
The DSC2 cells have both shorter APD and higher firing rate, and the shorter APD could be the result of not only increased K current but also the higher rate. This should be taken into consideration. Has the authors examined the mechanism of the higher rate and the ability of CA and SP in restoring normal rate? Have the authors measured the HCN channel/current?
It is apparent from the figure that CA and SP blocked/inhibited not only the sustained component of total K current but also the transient component. The contribution of blockade of Ito to the effects of the blockers should be examined/considered. The authors stated that CA and SP blocked IKr. To support this statement, the authors need to demonstrate the identity of the CA/SP sensitive current, rather than just showing the expression of the HERG and KCNQ1 genes.
CA and SP impacted and normalized both the cellular electrophysiological properties and myocyte mechanical properties. The authors postulated that RyR likely plays a role in mediating the effects of CA/SP on Ca dynamics. This should be examined experimentally. Do the compounds stimulate RyR activity and SR Ca release? If so, how does this effect actually result in reduced DADs?
Round 2
Reviewer 2 Report
Thank you for addressing all my comments.